# Three-Dimensional (3D) in vitro cell culture protocols to enhance glioblastoma research

**Janith Wanigasekara**[1,2,3,4]*, **Lara J. Carroll**[1], **Patrick J. Cullen**[1,5], **Brijesh Tiwari**[3], **James F. Curtin** [1,2,4]*

**1** BioPlasma Research Group, School of Food Science and Environmental Health, Technological University Dublin, Dublin, Ireland, **2** Environmental Sustainability & Health Institute (ESHI), Technological University Dublin, Dublin, Ireland, **3** Department of Food Biosciences, Teagasc Food Research Centre, Ashtown, Dublin, Ireland, **4** FOCAS Research Institute, Technological University Dublin, Dublin, Ireland, **5** University of Sydney, School of Chemical and Biomolecular Engineering, Sydney, Australia

* james.curtin@tudublin.ie (JFC); janith.manoharawanigasekara@tudublin.ie (JW)

**Data Availability Statement:** All relevant datasets that support the findings of this study are uploaded into OSF.io and can be accessed using the following DOI link: https://doi.org/10.17605/OSF.

## Abstract

Three-dimensional (3D) cell culture models can help bridge the gap between *in vitro* cell cultures and *in vivo* responses by more accurately simulating the natural *in vivo* environment, shape, tissue stiffness, stressors, gradients and cellular response while avoiding the costs and ethical concerns associated with animal models. The inclusion of the third dimension in 3D cell culture influences the spatial organization of cell surface receptors that interact with other cells and imposes physical restrictions on cells in compared to Two-dimensional (2D) cell cultures. Spheroids' distinctive cyto-architecture mimics *in vivo* cellular structure, gene expression, metabolism, proliferation, oxygenation, nutrition absorption, waste excretion, and drug uptake while preserving cell–extracellular matrix (ECM) connections and communication, hence influencing molecular processes and cellular phenotypes. This protocol describes the *in vitro* generation of tumourspheroids using the low attachment plate, hanging drop plate, and cellusponge natural scaffold based methods. The expected results from these protocols confirmed the ability of all these methods to create uniform tumourspheres.

## Introduction

Two-dimensional (2D) cell culture models have become a cornerstone of biological research due to its ease of use, cheap cost, and repeatability, however *in vivo* tissue complexity can only be reached utilizing Three-dimensional (3D) cell culture [1]. 3D cell cultures are an improved *in vitro* cell culture technology that uses an artificially produced microenvironment to grow cells in three dimensions. Cells in 3D cell culture contain natural cell-cell interactions as well as cell–extracellular matrix (ECM) component interactions, allowing them to proliferate *in vitro* in a microenvironment that closely reflects *in vivo* settings [2,3]. 3D cell culture is vital in drug testing, because it is capable of replacing both 2D cell culture and animal trials. The initial step of traditional drug development begins with 2D cell culture, followed by animal studies and clinical trials; around 95% of possible preclinical trials in all therapeutic areas fail to result in effective human treatments. The primary reason for this is that original data from 2D cell

IO/28CHK Three-Dimensional (3D) in vitro cell culture protocols to enhance glioblastoma research Hosted on the Open Science Framework doi.org.

**Funding:** Science Foundation Ireland, 17/CDA/4653, Professor Brijesh Tiwari (PI), Professor Patrick J. Cullen (Collaborator), Professor James F. Curtin (Collaborator) Teagasc, 2017228, Mr Janith Wanigasekara (Fellow) Science Foundation Ireland, 22/FFP-A/9189, Professor James Curtin (PI).

**Competing interests:** The authors have declared that no competing interests exist.

culture-based testing was deceptive and mispredicted cellular responses, resulting in enormous loss of time and resources and, eventually, delaying the identification of viable treatments [4,5]. 3D cell cultures are a simplified reductionist concept. When compared to a whole animal, it is very transparent and straightforward to mimic complicated processes like as growth, invasiveness, and toxicity [6]. Thus the 3D cell culture technology can be utilized to enhance the quality of laboratory experiments and minimizing overall expenditure. It will also be able to develop cancer models, for preclinical screening and monitoring, as well as novel *in vivo* cancer therapeutic research [5,7–9].

The main benefits of employing 3D cell culture for drug discovery include greater cell-cell contact, ECM-cell interactions, varied rates of cellular proliferation, oxygen and nutrition availability, physiological gradients for nutrients, waste, signalling factors and drugs, and the additional influence of stroma, all of which assist in replicating natural tissue distribution [1,5,10]. It can also simulate drug resistance, cellular microenvironment activity, and the expression of intrinsic and genetic variables [7,10]. The declared goal of the European REACH legislation is "To ensure a high level of protection of human health and the environment from the effects of hazardous chemicals. It strives for a balance: to increase our understanding of the possible hazards of chemicals, while at the same time avoiding unnecessary testing on animals" (European Chemicals Agency, 2020). 3D cell cultures complement the 3Rs principles of animal research (Replacement, Reduction, and Refinement) and REACH regulations by reducing the number of animals used in testing, as well as saving time, money, and ethical issues [1,10]. Animal testing is both costly and time consuming. Furthermore, if the animal is in pain or under stress throughout the experiment, it may alter the biochemical, physiological, and metabolic processes, which might misrepresent the efficacy and adverse effects of drugs [6,9,10]. 3D spheroid cultures can be readily established in many cell culture facilities using anchorage-independent or anchorage-dependent methods. Three methods that can be established in most cell culture facilities are outlined in detail here, low attachment plate method, hanging drop method and scaffold based method.

To produce spheroids, anchorage independent/scaffold-free approaches rely on non-adherent cell-to-cell aggregation. This anchorage independent category includes low attachment plate and hanging drop plate technologies [11]. Low attachment plates are culture plates with an ultra-low attachment hydrophilic polymer covering that promotes cell aggregation to create spheroids [2,12]. Cell adhesion to the culture surface is often mediated by ECM proteins such as collagen-I and fibronectin. The hydrophilic polymer covering prevents protein adsorption to the culture vessel surface, reducing monolayer cell adherence. Finally, low attachment plates stimulate cell aggregation via cell-cell and cell-ECM interactions, while limiting ECM interactions with the plastic surface [5]. The benefits of employing low adhesion plates are straightforward, simple, efficient spheroid formation improved repeatability, reproducibility and handling, suitable for multicellular spheroids / co-culture and the ability to grow a wide range of tumour cell types [3,13,14]. The disadvantage is that it is time consuming, lack of uniformity between spheroids, coated plates are expensive, continuous passage culture and toxicity analysis is difficult, that the success rate in long term passage is low, and not suitable for migration/invasion assays [4,9,15–17]. When considering hanging drop plates are open bottomless wells that encourage the development of droplets of media that offer room for the creation of spheroids by self-aggregation via gravity and surface tension [5,18]. Because there is no surface to connect to, cells develop inside a bubble of growth medium, and spheroids dangle in open bottomless wells, which are frequently contained at the bottom of the plate to adjust the cells' ambient humidity [12]. The well can typically accommodate up to 50μl of media while recommended drop volume is 10–20 μl [15] and the spheroid size is determined by the cell density [2]. This cell suspension droplet can be held in place by surface tension, and following 3D

spheroid creation, it may be dispensed by adding an extra drop of media to the well and the spheroid loaded to an another normal well plate [9]. The hanging drop plate method produce smaller size spheroids compared to low attachment plate method. The advantages of adopting the hanging drop plate method are the ability to create uniform size spheroids, the low cost, the ease of handling, the suitability for co-culturing, short term culture, and high throughput testing [3]. The fundamental disadvantage of this approach is that medium changes are difficult, different drug treatments at different time periods are impossible, it is not ideal for long-term culture, and it has a small culture volume [9,15].

Anchorage-dependent techniques employ pre-designed porous membranes and polymeric fabric meshes known as "scaffolds," which can be made of natural or synthetic materials [16]. This physical support can give structures ranging from basic mechanical to extracellular matrix-like structures [1]. 3D spheroids can be created by seeding cells into the 3D scaffold matrices or by distributing cells in a liquid matrix followed by solidification and polymerization. Cells are immersed in extracellular components and can begin cell-cell and cell-matrix interactions, as well as provide physical support for cell growth, adhesion, and proliferation [1]. Fibronectin, collagen, laminin, gelatin, cellulose, chitosan, glycosaminoglycans, fibroin, agarose, alginate, starch, and human decellularized ECM are some of the natural scaffolds [14,15,19,20]. The benefits of employing biological scaffolds are that they are extremely comparable to *in vivo* settings, that they can manage similar composition/elasticity/porosity to achieve better ECM presentation, and that they can be combined with appropriate growth factors. It can also increase biocompatibility and reduce toxicity. Disadvantages include the fact that it is an expensive, time-consuming, complicated procedure that is not ideal for large-scale manufacturing, that it is difficult to separate cells from scaffold for further investigations such as flow cytometry and confocal imaging [15]. Polymers such as polyglycolic acid, polylactic acid, polyorthoesters and aliphatic polyesters such as polycaprolactone (PCL), polystyrene (PS), polycaprolactone (PCL), polyethylene oxide (PEO), and polyethylene glycol (PEG) can be used to make synthetic scaffolds [1,14,15]. The advantages of using synthetic scaffold include the ability to control porosity, stiffness, elasticity, and permeability, higher versatility, reproducibility, enhanced workability, ease of use, and mechanical qualities of synthetic materials that can be adjusted according to the cell culture required, and also their chemical composition is well characterized [11,15]. The disadvantages include a lack of biodegradation, which may impair cellular function, inability to remove cells and perform cytotoxicity tests.

GBM Organoids are a novel experimental paradigm in the current reductionists' approach. The combination of embryonic stem cells, induced pluripotent stem cells or resident stem cells, as well as modern 3D culture, controlled environment and differentiation techniques has enabled us to enhance the self-organization capacity of pluripotent stem cells' to form human brain-like tissues known as brain organoids or mini-brains [21,22]. Brain organoids are a potential new technology that will open up new paths with the development of 3D cell culture for cancer modeling, ex vivo investigation of molecular and cellular pathways, develop personalized medicine, and the efficient discovery and development of therapeutics [22,23].

Ultimately, 3D models must have high-throughput application, easy and standardized culture protocols and analytic methodologies to get proper outcomes [1]. In the present study, we used three different methods to construct three different *in vitro* 3D glioblastoma tumour-spheroid models to closely mimic the natural *in vivo* environment, shape, and cellular response. This is the first time that we are reporting all the three different approaches for successful U-251MG, U-87 MG and A-172 human glioblastoma astrocytoma (GBM) tumoursphere development.

## Materials and methods

The part of the protocol described in this peer-reviewed article "U-251MG Spheroid generation using low attachment plate method" is published on protocols.io, https://dx.doi.org/10.17504/protocols.io.bszmnf46 and is included for printing as S1 File with this article. The part of the protocol described in this peer-reviewed article "U-251MG Spheroid Generation Using Hanging Drop Method Protocol" is published on protocols.io, https://dx.doi.org/10.17504/protocols.io.btstnnen and is included for printing as S2 File with this article. The part of the protocol described in this peer-reviewed article "U-251MG Spheroid generation using a scaffold based method protocol" https://dx.doi.org/10.17504/protocols.io.bszqnf5w is published on protocols.io, and is included for printing as S3 File with this article.

### Chemicals

All chemicals used in this study were supplied by Sigma-Aldrich—Merck Group unless stated otherwise.

### Ethics statement

The research project was approved by TU Dublin Research Ethics and Integrity Committee and involved the use of human samples. Human cancer cell lines (U-251 MG, U87 MG and A-172) were used in the study. These are cell lines obtained from reputable commercial cell banks, these are established, commercially available cell lines and consent was not obtained from the original donors. Animal tissue (fetal calf serum) was also used in the study. This was obtained from a reputable commercial company.

### 2D cell culture

The human glioblastoma multiforme cell line (U-251 MG, formerly known as U-373MG-CD14) was a gift from Michael Carty (Trinity College Dublin), U87 MG and A-172 human glioblastomas were purchased from an ATCC European Distributor (LGC Standards). The absence of mycoplasma was checked by using a MycoAlert PLUS Mycoplasma Detection kit (Lonza). Cells were maintained in Dulbecco's modified Eagle medium (DMEM)-high glucose supplemented with 10% fetal bovine serum (FBS) and 1% penicillin/streptomycin. Cells were maintained in a humidified incubator containing 5% $CO_2$ atmosphere at 37˚C in a TC flask T25, standard for adherent cells (Sarstedt). Cells were routinely sub-cultured when 80% confluence was reached using 0.25% w/v Trypsin-EDTA solution.

### Low attachment plate method for 3D cell culture

Protocols.io DOI: dx.doi.org/10.17504/protocols.io.bszmnf46.

U-251 MG, U-87 MG and A-172 human glioblastoma cells were used to generate tumour spheroids. Separately, the single cell suspensions were centrifuged at 5000 rpm for 5 min, removed the supernatant, tapped the tube and re-suspended the cell pellet in DMEM-high glucose supplemented with 10% FBS and 1% penicillin/streptomycin. The single-cell suspensions (with desired seeding density) were transferred to a sterile reservoir and seeded 200 μl/well into Nunclon™ Sphera™ 96-well low attachment plates (Thermo Fisher Scientific) using a multichannel pipette ensuring pipette tips do not touch the surface of the wells to protect the surface coating. The low attachment plates were centrifuged at 1250 rpm for 5 min followed by incubation (37˚C, 5% $CO_2$, 95% humidity). After 24h of incubation, the media must be replenished. 100 μl media was removed without disrupting the tumourspheres and 100 μl of fresh media (DMEM + 10%FBS + 1%Ab) was added into each well and incubate at 37˚C (5% $CO_2$,

95% humidity). The sides of wells should be used to remove or add media, and pipetting should be carried out at average or below average speeds to avoid disruption to spheroids. Tumour spheroid formation was observed within 4 days for U-251 MG, U-87 MG and A-172. Tumour spheroid formation was visually confirmed daily using an Optika XDS-2 trinocular inverse microscope equipped with a Camera ISH500, and their mean diameters were analyzed using "ImageJ version 1.53.e" software.

## Hanging drop plate method

Protocols.io DOI: dx.doi.org/10.17504/protocols.io.btstnnen.

U-251 MG, U-87 MG and A-172 single cell suspensions (with desired seeding densities) were used to generate tumour spheroids using HDP1096 Perfecta3D® 96-Well Hanging Drop Plate. Sterile PBS was added to the reservoirs located on the peripheral rims, which are divided into sections. 2 ml of PBS was added to each plate reservoir section, and 1ml was added per tray reservoir section. This prevented the hanging drop from drying throughout the incubation period. In order to achieve 5000 cells per 20 μl of hanging drop, the single cell suspensions prepared in DMEM-high glucose supplemented with 10% FBS and 1% penicillin/streptomycin at a concentration of $2.5x10^5$ cells/ml. Each hanging drop well was able to hold 20–50 μl of cell suspension, and any volume above 50 μl resulted in droplet instability. Hanging drops can be formed by carefully pipetting 20–50 μl of cell suspension into the centre of each well from the top side of the plate. Hanging drops should be formed on and confined to the bottom of the plate. Placed the lid on the plate and inserted the assembly into a humidified cell culture incubator at 37˚C and 5% $CO_2$. Tumour spheroids formation was visually confirmed within 4 days for U-251 MG, U-87 MG and A-172. 5μl of fresh media was added back into the hanging drops by placing the end of the pipette tips in the neck region of the access holes/wells and the fresh media was slowly dispensed into the access holes. Once formed, tumourspheres can be transferred from the hanging drop plate to low attachment plates/pre-coated wells in the dish by adding 50 μl of fresh media into each hole.

## Scaffold based method

Protocols.io DOI: dx.doi.org/10.17504/protocols.io.bszqnf5w.

U-251 MG, U-87 MG and A-172 single cell suspensions (with desired seeding densities) were used to generate tumour spheroids using cellusponge 3D scaffolds. A 9 mm cellusponge disk was slowly placed in the middle of each well in a 24-well plate and 100 μL was seeded from a cell suspension with a cell density of 5000k cells/ml. Cellusponge disks with cells were incubated at 37˚C (5% $CO_2$, 95% humidity) for 3 hours to remove any air bubbles within the cellusponge. After incubation, 500 μL of DMEM-high glucose supplemented with 10% FBS and 1% penicillin/streptomycin was added slowly along the edge of each well in a 24-well plate. Plates with cellusponge scaffolds were incubated overnight at 37˚C, 5% $CO_2$, 95% humidity. After overnight incubation, the seeded scaffolds were transferred into a new well plate, the media was replenished and the culture should be continued. Tumour spheroid formation was observed within 3–4 days.

## Image J analysis

Tumour spheroid formation was visually confirmed daily using an Optika XDS-2 trinocular inverse microscope equipped with a Camera ISH500, and their mean diameters were analysed using "ImageJ version 1.53.e" software (http://imagej.nih.gov/ij/). ImageJ is a free software that can be used for manually counting the cell numbers and calculating the cellular size (area / diameter). The ImageJ program was calibrated (set scale) using an image obtained from the

same microscope with a known scale before it was used to calculate the cell size (in diameter). Following the calibration, the pictures of the tumourspheres were opened in the program, and a line was drawn across the diameter to measure the tumoursphere's size. The diameters of the spheroids were measured at least three times to obtain the mean and standard deviation.

## Growth analysis at different incubations

The growth of U-251 MG, U-87 MG and A-172 tumourspheres were analysed during different incubations (ranging from 24 to 168 h). Cells (initial seeding density was 10000 cells/ml) were seeded in the above mentioned Nunclon™Sphera™96-well-low attachment plates. Fresh media were added every third day by replenishing old media in each well without disturbing the tumourspheroids. In the hanging drop plate method, 5000 cells/well were seeded in the HDP1096 Perfecta3D® 96-well Plate. While in scaffold based method, 5000k cells/ml were seeded in the hydroxipropylcellulose scaffold. The spheroid formation and growth were monitored daily by using an inverted phase-contrast microscope, and the sizes of the spheroids were measured as described above using at least nine spheroids within the three biological repetitions.

## Growth analysis at different seeding densities

For growth analysis, varying numbers of U-251 MG, U-87 MG and A-172 cells (ranging from 2000 to 40 000 cells/ml) were seeded in the above mentioned Nunclon™ Sphera™ 96-well-low attachment plates for 96 hours. Fresh media were added every third day by replenishing old media in each well without disturbing the tumourspheroids. In the hanging drop plate method, U-251 MG cells (ranging from 1000 to 10000 cells/well) were seeded in the above mentioned HDP1096 Perfecta3D® 96-well Plate. While in scaffold based method, varying numbers of U-251 MG cells (ranging from $1x10^6$ to $6x10^6$ cells/ml) were seeded in the hydroxipropylcellulose scaffold. The spheroid formation was monitored after 96 h by using an inverted phase-contrast microscope, and the sizes of the spheroids were measured as explained above, using at least nine spheroids within the three independent experiments.

## Spheroid cells health analysis

Spheroid cell health was analysed using the Alamar Blue™ cell viability reagent (Thermo Fisher Scientific). After the post treatment incubation, tumourspheres were washed with sterile phosphate-buffered saline (PBS), trypsinized using a 0.25% w/v trypsin–EDTA solution and incubated for 3 h at 37˚C with a 10% Alamar Blue™ solution [10]. During the scaffold based method tumourspheres embedded in the cellusponge 3D scaffolds were incubated for 24h instead of 3h [24]. Fluorescence was measured using an excitation wavelength of 530 nm and an emission wavelength of 590 nm with a Varioskan Lux multiplate reader (Thermo Scientific). The fluorescence signals were normalized by spheroid size (in diameter); a higher ratio indicates healthier spheroids. The experiments consisted of three independent tests in which at least nine spheroids were measured throughout three biological repeats.

## CellTiter-Glo® 3D cell viability assay

3D cell viability was analysed using the CellTiter-Glo® 3D Cell viability assay (Promega). After the post treatment incubation, homogeneous tumourspheres were removed from the 96-well low attachment culture plate and placed separately in single wells of a 96-well plate (Sarstedt). CellTiter-Glo® 3D reagent was added to each well and the luminescence signals were read after 25 minutes of incubation at room temperature using the Varioskan Lux multiplate reader (Thermo Scientific).

## Temozolomide induced glioma cytotoxicity

Dose response curves for the commonly employed chemotherapeutic drug, Temozolomide (TMZ), used for the treatment of U-251 MG, U-87 MG, and A-172 GBM tumour spheroids. TMZ was dissolved in dimethyl sulfoxide (DMSO) and stored at −20˚C. These stocks were subsequently used to make the working standard solutions in media. The highest concentration of DMSO used was 0.5%. U-251 MG, U-87 MG and A-172 cells were seeded at a density of $1 \times 10^4$ cells/ml (200 μl culture medium/well) into Nunclon™ Sphera™ 96-well low attachment plates (Thermo Fisher Scientific). After tumoursphere construction, existing media were removed from each well and tumourspheres were treated with TMZ (concentration gradient from 500 μM to 0.97 μM), and incubated for 6 days at 37˚C (5% $CO_2$, 95% humidity). DMSO (20%) was used as a positive control. After the post treatment incubation, the cytotoxicity of the tumourspheres were measured using the both CellTiter-Glo® 3D Cell viability assay and Alamar Blue™ cell viability reagents as mentioned above, using at least three independent tests with a minimum of three replicates per experiment.

## Statistical analysis

All the experiments were replicated at least three independent times. Prism versions 9.1.0, GraphPad Softwares, Inc. were used to carry out curve fitting and statistical analysis. Dose-response curves were measured using nonlinear regression. Data are presented as a percentage and error bars of all figures were presented using the standard error of the mean (SEM), multiple comparison analyzes were performed using Tukey's test unless otherwise stated.

## Expected results and discussion

GBM is distinguished by increased vascularization, significant cell heterogeneity, self-renewing cancer stem cells and the interactions between tumour and microenvironment, all of which contribute to tumour progression [25]. Tumour development, metastasis, angiogenesis, cytotoxicity resistance, and immune cell modulation are all influenced by the tumour microenvironment (TME) [10,26]. There is a gap in mostly accessible GBM pre-clinical models and 3D cell culture is able to fill this gap by providing more reliable models to study the correlation between TME, tumour reoccurrence and therapy resistance.

Three distinct approaches, such as low attachment plate (Fig 1-II), (S1 File), hanging drop plate (Fig 1-III), (S2 File), and scaffold based methods (Fig 1-IV), (S3 File) were used to create U-251MG, U-87 MG and A-172 3D human glioblastoma cell culture models. This facilitated 3D cell–cell and cell–ECM interactions and mirrored the diffusion-limited distribution of oxygen, nutrients, metabolites, and signaling molecules seen in the microenvironment of *in vivo* tumours [10]. Most research to date has used 2D cell culture (Fig 1-I), which has limitations as experimental models to predict biological responses, as explained previously.

U-251 MG, U-87 MG, and A-172 human glioblastoma astrocytoma spheroids formation and growth were monitored daily by using an inverted phase-contrast microscope, and their mean diameters were analysed using "ImageJ version1.53.e" software for at least three independent experiments. U-251MG tumoursphere growth during the low attachment plate method was found to be significantly increased with the incubation time, the size ranging from 135 μm, 229 μm, 323 μm and 461 μm (Fig 2A) for 24 to 96 h incubation respectively. U-87 MG tumourspheres were significantly increased with the incubation time, the size range from 129 μm, 234 μm, 303 μm and 357 μm (Fig 2B) for 24 to 96 h incubation respectively. While, A-172 tumourspheres also showed same behaviour with the increasing incubation time and the sizes rage from 71 μm, 191 μm, 240 μm and 367 μm (Fig 2C) for 24 to 96 h incubation respectively.

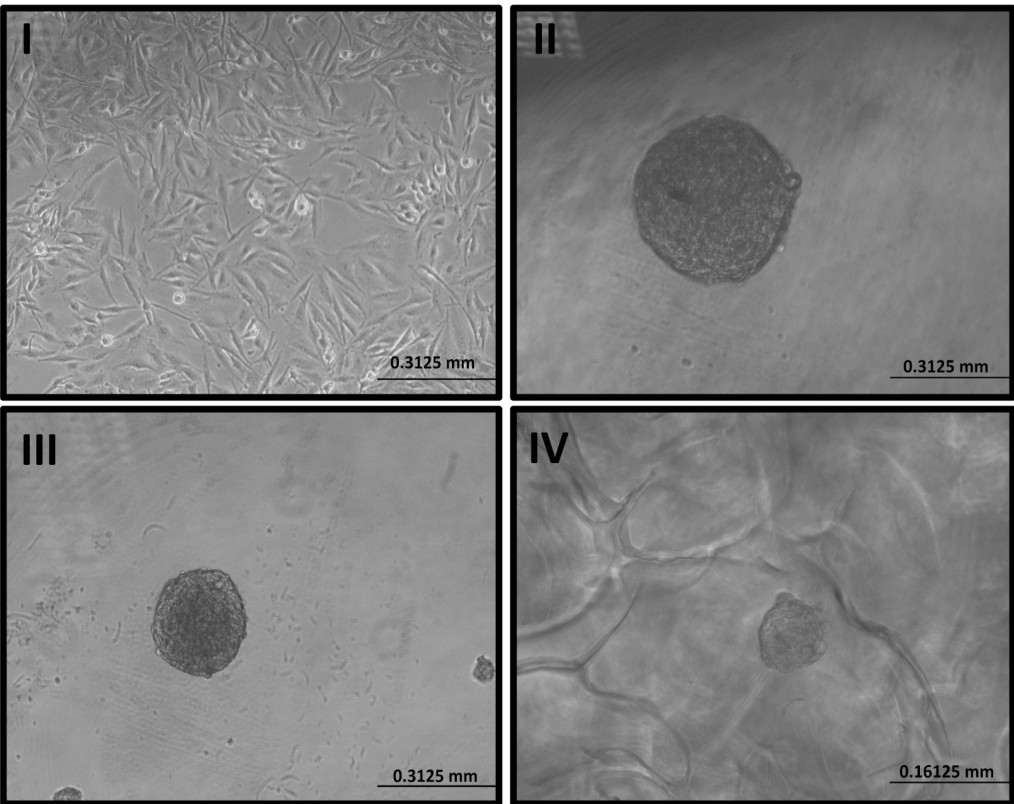

**Fig 1. Development of U-251 MG human glioblastoma astrocytoma 3D *in vitro* cell culture model.** I). Image of U-251 MG 2D cells in T75 flask II) 3D tumoursphere constructed in low adhesion plate. III). 3D tumoursphere constructed in hanging drop plate. IV) 3D tumoursphere constructed in hydroxipropylcellulose scaffold. Tumour spheroid formation was visually confirmed using an Optika XDS-2 trinocular inverse microscope equipped with a Camera ISH500.

U-251 MG tumoursphere growth during the hanging drop plate method was shown to be considerably enhanced with the incubation time, the size ranging from 105 μm, 139 μm, 208 μm and 269 μm (Fig 2D) for 24 to 96 h incubation respectively. U-87 MG tumourspheres were significantly increased with the incubation time, the size range from 92 μm, 143 μm, 224 μm and 252 μm (Fig 2E) for 24 to 96 h incubation respectively. While, A-172 tumourspheres also showed same behaviour with the increasing incubation time and the sizes rage from 63 μm, 131 μm, 207 μm and 265 μm (Fig 2F) for 24 to 96 h incubation, respectively.

U-251 MG tumoursphere growth in hydroxipropylcellulose 3D scaffold was shown to be considerably enhanced with incubation time, with sizes ranging from 22 μm, 49 μm, 70 μm, and 110 μm for 24 to 96 h incubation, respectively (Fig 2G). U-87 MG tumourspheres were significantly increased with the incubation time, the size range from 28 μm, 55 μm, 90 μm and 143 μm (Fig 2H) for 24 to 96 h incubation respectively. While, A-172 tumourspheres size also significantly enhanced with the increasing incubation time and the sizes rage from 17 μm, 51 μm, 69 μm and 97 μm (Fig 2I) for 24 to 96 h incubation respectively.

These results proved that these protocols have the ability to develop 3D tumourspheres and the presence of heterogeneous cellular subpopulations such as actively proliferating, quiescent, hypoxic, and necrotic cells [2,14,27].

The optimum U-251 MG (Fig 3A), U-87 MG (Fig 3B) and A-172 (Fig 3C) tumourspheroids formations were observed within 96 h of incubation for the 10000 cells/ml initial seeding

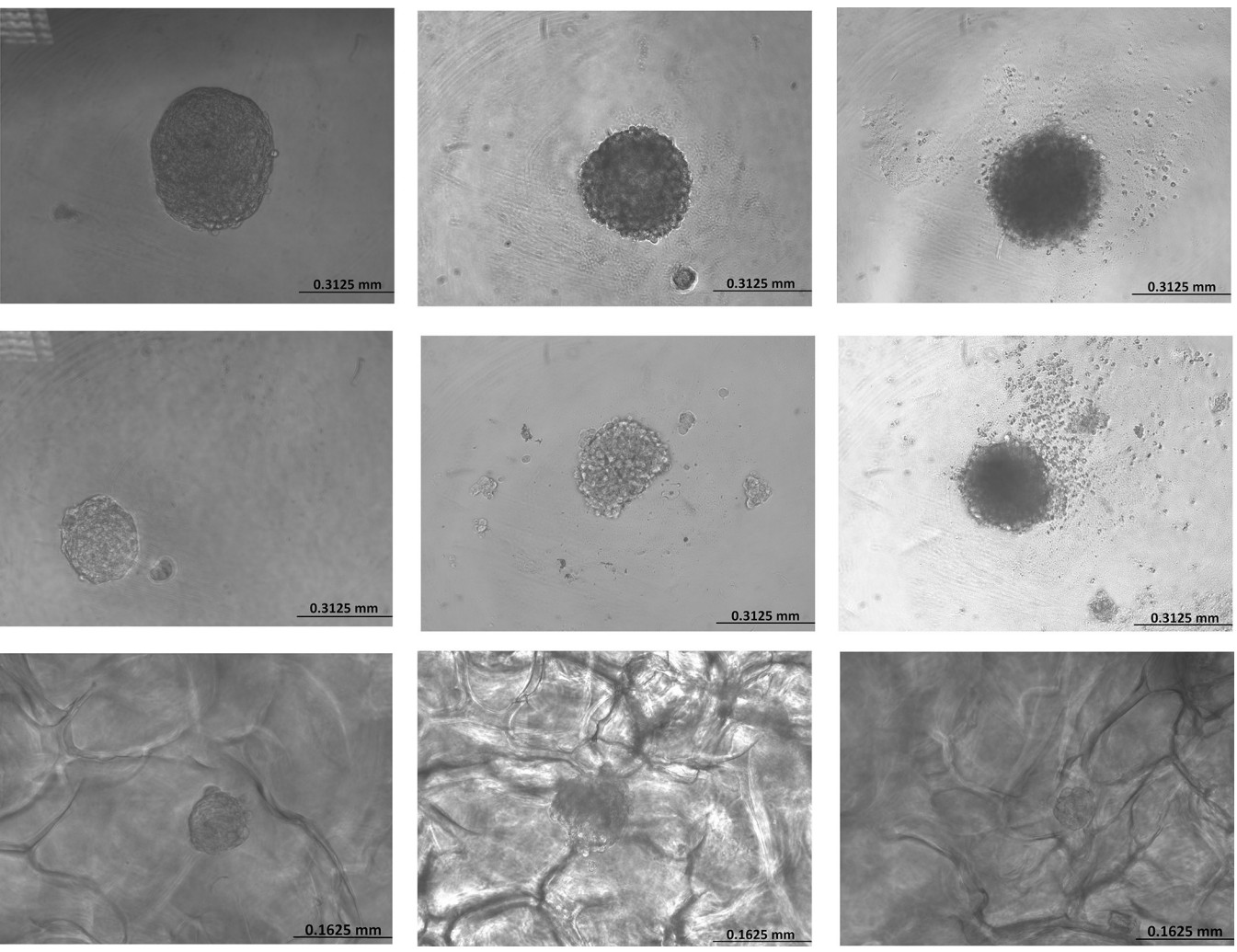

**Fig 2. Development of U-251 MG, U-87 and A-172 human glioblastoma astrocytoma 3D *in vitro* cell culture models using low attachment plate, hanging drop plate and scaffold based method.** A). U-251 MG, B) U-87 MG, C) A-172 tumourspheroids formation after 96 h of incubation using low attachment plate method, D) U-251 MG, E) U-87 MG, F) A-172 tumourspheroids formation after 96 h of incubation using hanging drop plate method, G) U-251 MG, H) U-87 MG, I) A-172 tumourspheroids formation after 96h of incubation using scaffold based method.

density. One-way analysis of variance (ANOVA) demonstrated that there were significant differences in tumourspheres diameter during 24–96 h incubation, while there was no significant difference during 96–168 h incubation. It was also observed that exponential growth (Log) was achieved within the initial 4 days of growth, after which the growth curve became stationary in all these three glioblastoma cell lines.

For growth analysis, varying numbers of U-251 MG, U-87 MG and A-172 cells (ranging from 2000 to 40 000 cells/ml) were seeded in the Nunclon™ Sphera™ 96-well low attachment plates as explained above. The largest U-251 MG tumourspheres were observed with 10 000, 15 000, and 20 000 cells/ml initial seeding densities after 96 h of incubation. One-way ANOVA demonstrated that there was a significant difference in tumoursphere diameter between each initial seeding densities as shown in Fig 3D. However, there was no significant difference between diameters in 10 000, 15 000, and 20 000 cells/ml seeding densities. The largest U-87 MG tumourspheres were observed with 40 000 cells/ml initial seeding densities after 96 h incubation. One-way ANOVA demonstrated that there was a significant difference in

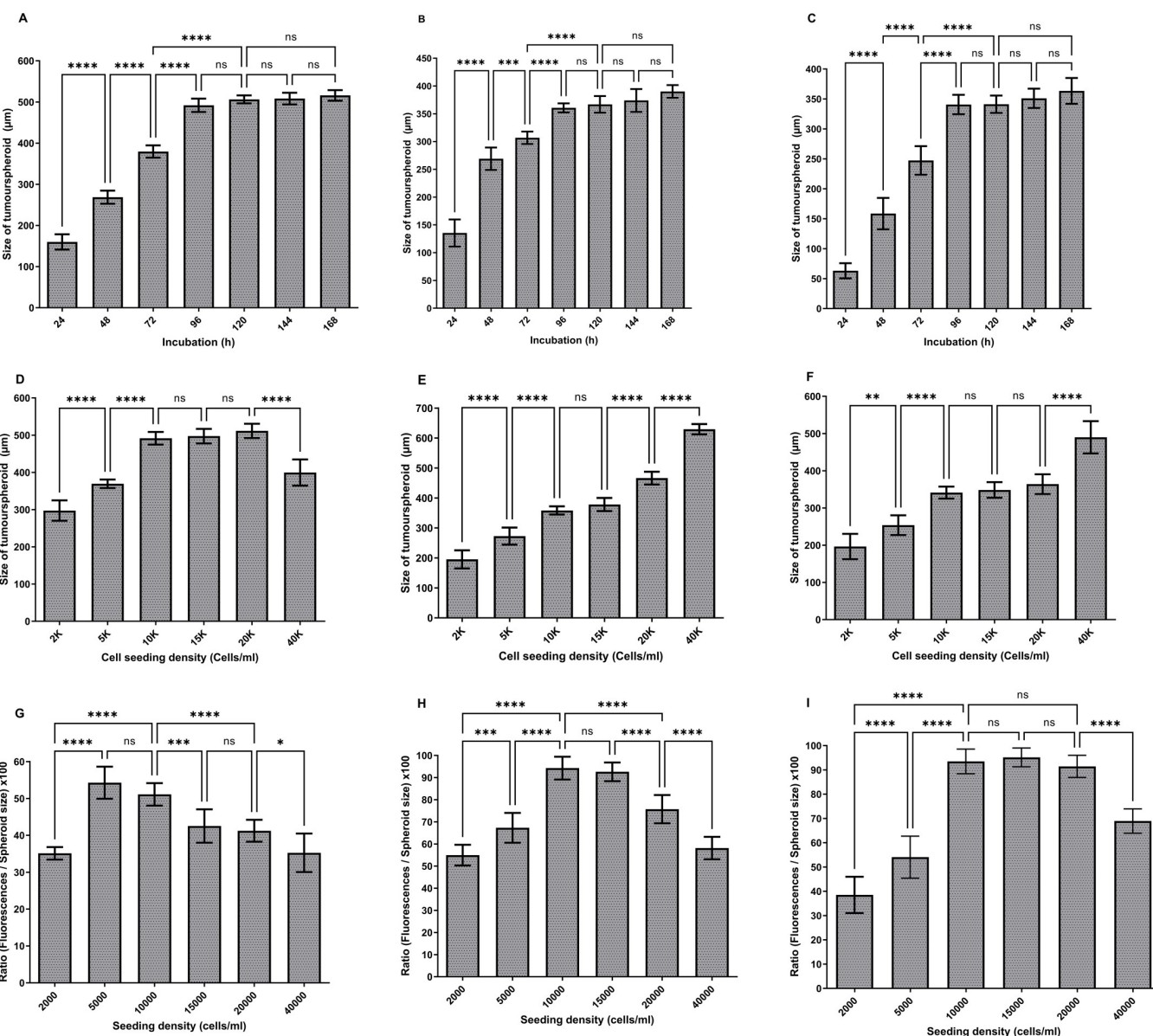

**Fig 3. Development of U-251 MG human glioblastoma astrocytoma 3D *in vitro* cell culture models using low attachment plate method.** A) U-251 MG, B) U-87 MG, C) A-172 tumourspheres growth (diameter in μm) analysis during different incubations. D) U-251 MG E) U-87 MG, F) A-172 growth analysis after 96h incubation (diameter in μm) at increasing seeding density. The mean of the diameter was used to plot the values on columns and analysed using one-way ANOVA with tukey's post-test (ns, not significant ($p > 0.05$); *$p < 0.05$; **$p < 0.01$, ***$p < 0.001$; ****$p < 0.0001$). G) U-251 MG H) U-87 MG, I) A-172 spheroid cell health analysed and a higher ratio indicates healthier spheroids. The mean of the [(fluorescence / spheroid size) x 100] was used to plot the values on columns and analysed using one-way ANOVA with tukey's post-test (ns, not significant ($p > 0.05$); *$p < 0.05$; **$p < 0.01$, ***$p < 0.001$; ****$p < 0.0001$).

tumoursphere diameter between each of the initial seeding densities as shown in Fig 3E. However, there was no significant difference between diameters at 10 000 and 15 000 cells/ml seeding densities. While, the largest A-172 tumourspheres were observed with 40 000 cells/ml initial seeding densities after 96h incubation. One-way ANOVA demonstrated that there was a significant difference in tumoursphere diameter between each of the initial seeding densities as shown in Fig 3F. Though, there was no significant difference between diameters in 10 000, 15 000, and 20 000 cells/ml seeding densities.

U-251 MG, U-87 MG and A-172 cells health analysed after 96h incubation using Alamar Blue™ cell viability reagent as explained above and the fluorescence signals were normalized by spheroid size (diameter in μm). A higher ratio suggests that the spheroids are healthier. During U-251 MG growth confirmed that 5000 and 10 000 cells/ml initial seeding densities were having highest spheroids cell health. One-way ANOVA confirmed that there was no significant difference in tumoursphere health during 5000 and 10 000 cells/ml (Fig 3G). During U-87 MG growth, it was confirmed that 10 000 and 15 000 cells/ml initial seeding densities were having highest spheroids cell health. One-way ANOVA confirmed that there was no significant difference in tumoursphere health during 10 000 and 15 000 cells/ml (Fig 3H). During A-172 growth confirmed that 10 000, 15 000 and 20 000 cells/ml initial seeding densities were the ones having the highest spheroids cell health. One-way ANOVA confirmed that there was no significant difference in tumoursphere health during 10 000, 15 000 and 20 000 cells/ml (Fig 3I).

We studied tumoursphere growth in low attachment plates with various seeding densities and observed tumoursphere growth ranging in diameter from 150 to 650 μm. Our findings are correlate with the tumoursphere diameters determined by Singh et al [28]. According to the results, 10000 cells/ml initial seeding density was the most suitable seeding density for low attachment plate method and all the above glioblastoma cell lines were able to produce healthy tumourspheres after 96h incubation. Recently, we used the same protocol to generate U-251MG tumourspheres and successfully studied the plasma induced cytotoxicity in 3D glioblastoma tumour spheroids [10].

This results also proved low attachment plate's ability to promote aggregation of cells by cell-cell and cell-ECM interactions while blocking the ECM interaction to the plastic surface. Which can be used as a pre-clinical model due to its simplicity, efficiency, higher reproducibility and also possible to generate a wide range of tumour cell types using the same protocol in any laboratory conditions [9,15,16,29].

The optimum U-251 MG (Fig 4A), U-87 MG (Fig 4B) and A-172 (Fig 4C) tumourspheroids formations were attained after 96 h of incubation for the 5000 cells/well initial seeding density by achieving a size range of 251–285 μm, 252–279 μm and 217–265 μm respectively. One-way ANOVA indicated that there were significant differences in tumourspheres diameter during 24–96 h incubation, while, there were no significant difference during 96–168 h incubation. It was also observed that exponential growth (Log) was achieved within the initial 4 days of growth, after which the growth curve became stationary in all these three glioblastoma cell lines.

For growth analysis, varying numbers of U-251 MG, U-87 MG and A-172 cells (ranging from 1000 to 10 000 cells/well) were seeded in the HDP1096 Perfecta3D® 96-well hanging drop plates and the mean sizes were computed after 96h of incubation. The largest U-251 MG tumourspheres were observed with 10 000 cells/well initial seeding densities after 96 h incubation. As illustrated in Fig 4D, one-way ANOVA revealed a significant difference in tumoursphere diameter between each initial seeding density. The largest U-87 MG tumourspheres were observed with 8 000 to 10 000 cells/well initial seeding densities after 96 h of incubation. One-way ANOVA demonstrated that there is a significant difference in tumoursphere diameter between each of the initial seeding densities as shown in Fig 4E. However, there was no significant difference between diameters in 5 000, 8 000 and 10 000 cells/well seeding densities. While the largest A-172 tumourspheres were observed with 8 000 to 10 000 cells/well initial seeding densities after 96 h of incubation. One-way ANOVA demonstrated that there is a significant difference in tumoursphere diameter between each of the initial seeding densities as shown in Fig 4F. Though, there was no significant difference between diameters in 4 000 to 5 000, and 8 000 to 10 000 cells/well seeding densities.

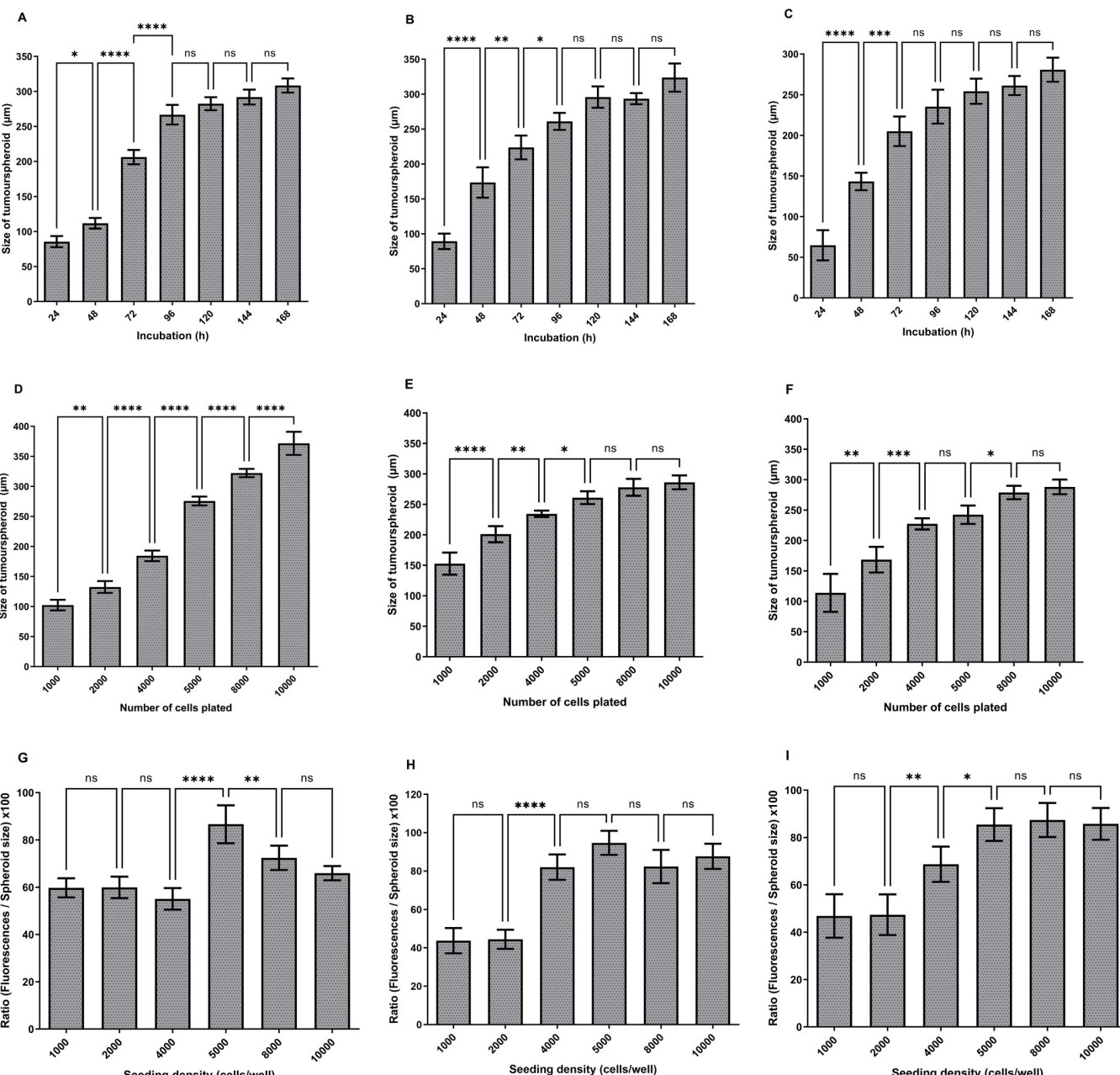

**Fig 4. Development of U-251 MG human glioblastoma astrocytoma 3D *in vitro* cell culture model using hanging drop plate method.** A) U-251 MG, B) U-87 MG, C) A-172 tumourspheres growth (diameter in μm) analysis during different incubations. D) U-251 MG E) U-87 MG, F) A-172 growth analysis after 96h incubation (diameter in μm) at increasing seeding density. The mean of the diameter was used to plot the values on columns and analysed using one-way ANOVA with tukey's post-test (ns, not significant ($p > 0.05$); *$p < 0.05$; **$p < 0.01$, ***$p < 0.001$; ****$p < 0.0001$). G) U-251 MG H) U-87 MG, I) A-172 spheroid cell health analysed and a higher ratio indicates healthier spheroids. The mean of the [(fluorescence / spheroid size) x 100] was used to plot the values on columns and analysed using one-way ANOVA with tukey's post-test (ns, not significant ($p > 0.05$); *$p < 0.05$; **$p < 0.01$, ***$p < 0.001$; ****$p < 0.0001$).

During the U-251 MG spheroids cell health investigation, it was established that the initial seeding density of 5000 cells/well had the best spheroids cell health. The substantial difference in 4000 to 5000 cells/well and 5000 to 8000 cells/well was verified by one-way ANOVA, however there was no significant difference in tumoursphere health at the other seeding densities (Fig 4G).

U-87 MG growth confirmed that 5 000 cells/well initial seeding density was having highest spheroids cell health. One-way ANOVA confirmed that there was no significant difference in tumoursphere health during 4 000, 5 000, 8 000 and 10 000 cells/well (Fig 4H). During A-172 growth confirmed that 5 000, 8 000 and 10 000 cells/well initial seeding densities were having highest spheroids cell health. One-way ANOVA confirmed that there was no significant difference in tumoursphere health during 5 000, 8 000 and 10 000 cells/well (Fig 4I).

We studied tumoursphere growth in hanging drop plates with various seeding densities and observed tumoursphere growth ranging in diameter from 100 to 400 μm. According to the results, 5000 cells/well initial seeding density was the most suitable seeding density for the hanging drop plate method, and all the above glioblastoma cell lines were able to produce healthy tumourspheres after 96 h incubation. This result also proved the hanging drop plate's ability to produce uniform sized spheroids, ability to control the size of spheroid by seeding density, higher replicability, lower cost, and ability of tumoursphere mass production within a shorter time period [9,12,15,29].

The largest U-251 MG and A-172 tumourspheroids formation were attained after 120 h incubation by achieving a size range 110–156 μm (Fig 5A) and 146–174 μm (Fig 5C) for the 5000k cells/ml initial seeding density respectively. One-way ANOVA indicated that there were significant difference in tumoursphere diameter throughout the incubation. The optimum U-87 MG tumourspheroid formation was observed within 120 h of incubation (size range from 133–191 μm) for the 5000k cells/ml initial seeding density. One-way ANOVA indicated that there were significant differences in tumourspheres diameter during 24–120 h incubation, while, there was no significant difference during 48–72 h incubation (Fig 5B).

For growth analysis, varying numbers of U-251 MG, U-87 MG and A-172 (ranging from $1\times10^6$ to $6\times10^6$ cells/ml) were seeded in the hydroxipropylcellulose 3D scaffolds. Fresh media were added every third day by replenishing old media in each well without disturbing the scaffolds, and the mean sizes were calculated after 120 h of incubation. The largest U-251 MG tumourspheres were detected with $5\times10^6$ and $6\times10^6$ cells/ml initial seeding densities after 120 h of incubation. One-way ANOVA verified that there is a significant difference in tumoursphere diameter between $4\times10^6$ and $5\times10^6$ seeding densities, while there was no significant difference in diameters between $5\times10^6$ and $6\times10^6$ cells/ml initial seeding densities as shown in Fig 5D. The largest U-87 MG and A-172 tumourspheres were observed with $5\times10^6$ and $6\times10^6$ cells/ml initial seeding densities after 120 h of incubation. One-way ANOVA demonstrated that there were significant difference in tumoursphere diameter between each initial seeding densities as shown in Fig 5E and 5F respectively. However, there was no significant difference between diameters in $5\times10^6$ and $6\times10^6$ cells/ml seeding densities.

U-251 MG, U-87 MG and A-172 spheroids cell health were analysed after 120 h of incubation as explained above, U-251 MG growth confirmed that $5\times10^6$ cells/ml initial seeding density was having highest spheroids cell health. One-way ANOVA confirmed that there were significant difference in tumoursphere health during $4\times10^6$, $5\times10^6$ and $6\times10^6$ cells/ml. While there was a significant difference between $3\times10^6$ and $4\times10^6$ densities as shown in Fig 5G.

During U-87 MG growth, it was confirmed that $5\times10^6$ cells/ml initial seeding densities were having the highest spheroids cell health. One-way ANOVA confirmed that there were significant differences in tumourspheres health from $1\times10^6$ to $6\times10^6$ cells/ml (Fig 5H). During A-172 growth, it was confirmed that $5\times10^6$ and $6\times10^6$ cells/ml initial seeding densities had the highest spheroids cell health. One-way ANOVA confirmed that there was no significant difference in tumoursphere health during $4\times10^6$, $5\times10^6$ and $6\times10^6$ cells/ml (Fig 5I).

We studied tumoursphere growth in hydroxipropylcellulose 3D scaffolds with various seeding densities and observed tumoursphere growth ranging in diameter from 50 to 200 μm. According to the results, $5\times10^6$ cells/well initial seeding density was the most suitable seeding density for the

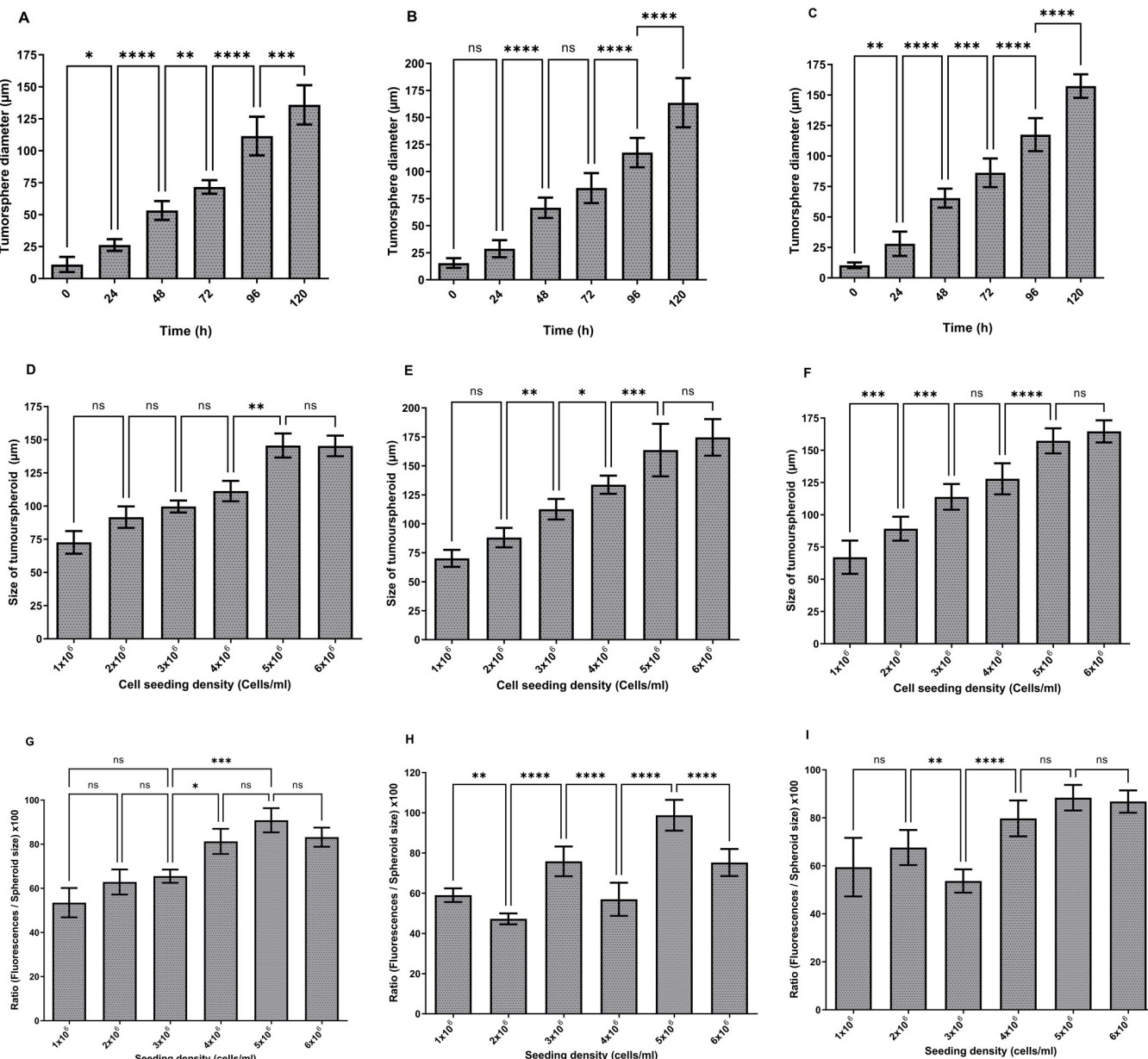

**Fig 5. Development of U-251 MG human glioblastoma astrocytoma 3D *in vitro* cell culture model using cellusponge 3D scaffolds.** A) U-251 MG, B) U-87 MG, C) A-172 tumourspheres growth (diameter in μm) analysis during different incubations. D) U-251 MG E) U-87 MG, F) A-172 Growth analysis after 120 h incubation (diameter in μm) at increasing seeding density. The mean of the diameter was used to plot the values on columns and analysed using one-way ANOVA with tukey's post-test (ns, not significant (p > 0.05); *p < 0.05; **p < 0.01, ***p < 0.001; ****p < 0.0001). G) U-251 MG H) U-87 MG, I) A-172 spheroid cell health analysed and a higher ratio indicates healthier spheroids. The mean of the [(fluorescence / spheroid size) x 100] was used to plot the values on columns and analysed using one-way ANOVA with tukey's post-test (ns, not significant (p > 0.05); *p < 0.05; **p < 0.01, ***p < 0.001; ****p < 0.0001).

scaffold based method and all the above glioblastoma cell lines were able to produce healthy tumourspheres after 120 h of incubation. These results also proved the biological scaffold's higher biocompatibility and its possibility to control scaffold's composition, porosity, and elasticity to get better GBM ECM representation [15,20]. This protocol can be applied to hydroxipropylcellulose scaffolds or any other natural scaffold [30] and possible to generate wide range of tumoursphere types in any laboratory. It is also possible to improve scaffold chemistry and composition to mimic the physiological architecture of any glioblastoma tumours.

**A**

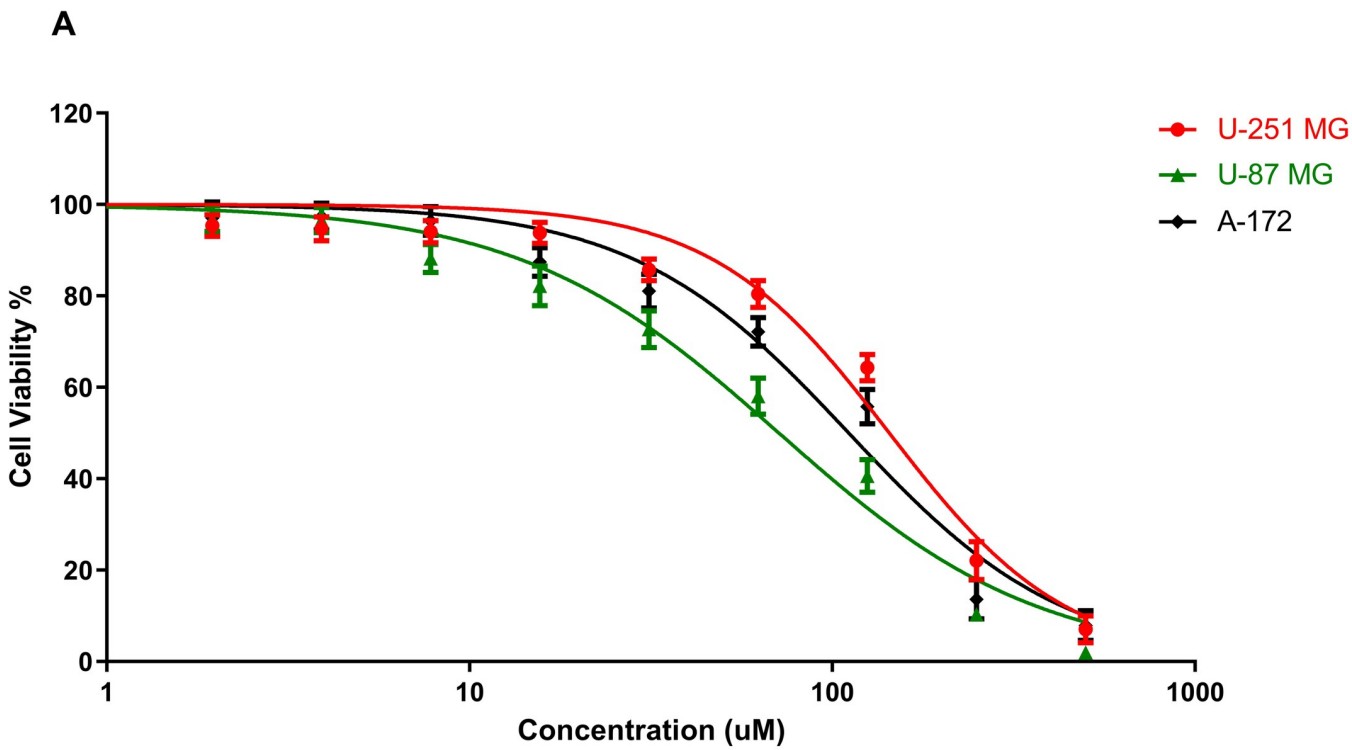

**B**

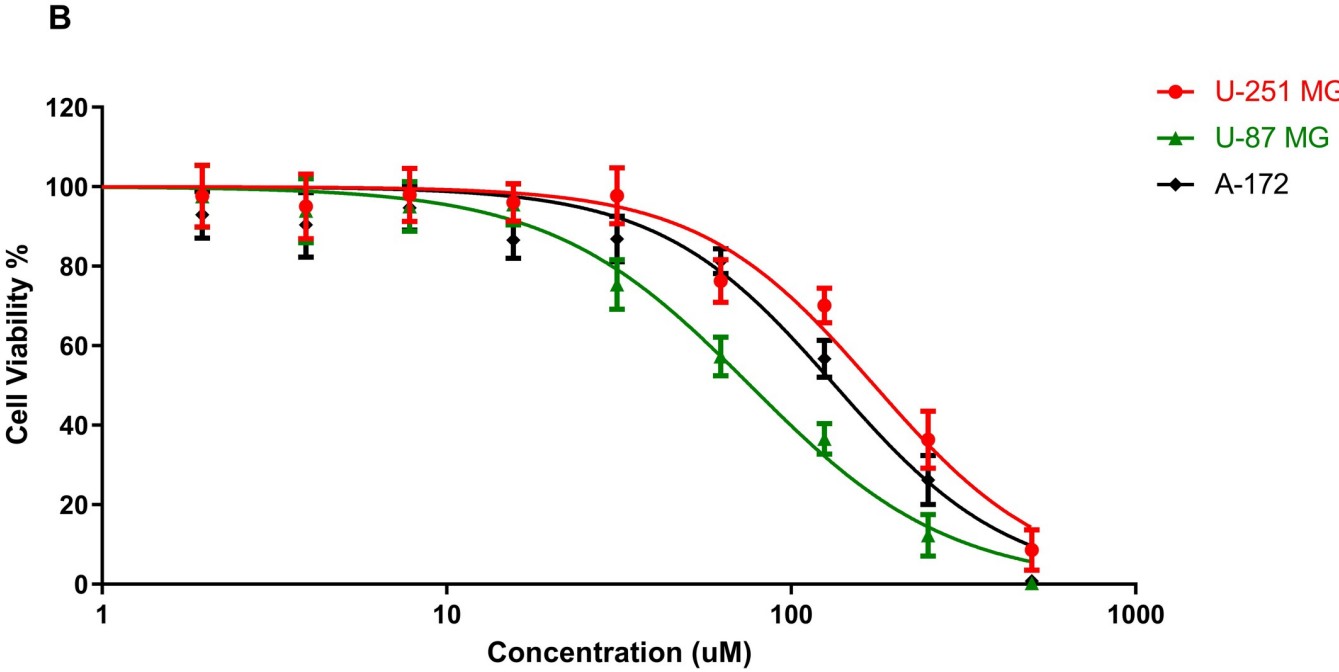

**Fig 6. TMZ induced cytotoxicity in U-251 MG, U-87 MG and A-172 tumourspheres with 6 days post treatment incubation.** A) Cytotoxicity analysis by using Alamar Blue™ cell viability assay B) CellTiter-Glo® 3D Cell viability assay.

The effects of TMZ cytotoxicity on the different GBM cell lines were studied using U-251 MG, U-87 MG and A-172 tumourspheres. TMZ induced cytotoxicity was studied using two different cell viability assays as shown in Fig 6. The CellTiter-Glo® 3D Cell viability assay quantifies the amount of ATP present, which is a marker for the presence of metabolically active cells, to determine the number of viable cells in a 3D cell culture. While alamarBlue cell viability, Resazurin is used as an oxidation-reduction (REDOX) indicator that undergoes colorimetric change in response to cellular metabolic reduction.

TMZ treated tumourspheres (concentration gradient from 500 µM to 0.97 µM), were post incubated for 6 days at 37˚C. An $IC_{50}$ of 143.6 µM (135.2 ± 152.6 µM), 71.25 µM (66.41 ± 76.44 µM), and 111.0 µM (103.1 ± 119.4 µM) were found for U-251 MG, U-87 MG and A-172 tumourspheres respectively, when analysed by using the Alamar Blue™ cell viability assay (Fig 6A). An $IC_{50}$ of 174.4 µM (160.5 ± 189.5 µM), 76.06 µM (70.81 ± 81.70 µM), and 134.0 µM (115.5 ± 155.3 µM) were found for U-251 MG, U-87 MG and A-172 tumourspheres respectively, when analysed by using the CellTiter-Glo® 3D Cell viability assay (Fig 6B). Two-way ANOVA demonstrated that there were significant differences in viability between the different TMZ concentrations and different cell lines ($p < 0.0001$). According to these results, it was postulated that as U-87 MG has the highest TMZ sensitivity, while U-251 MG tumourspheres showed highest cell viability with TMZ treatment. TMZ induced cytotoxicity in all three cell lines showed similar values, when comparing both cell viability assays. However, comparatively higher IC50 values were observed from the CellTiter-Glo® 3D Cell viability assay, and this may be due to the difference in assay chemistries and metabolic targets in viability assays.

The effects of diffusion of the active dyes through the matrices and their subsequent bio-availability to the cells can lead to misinterpretation of the results obtained. The concern is addressed in the present study by converting tumourspheres into single cells before cell viability analysis using the Alamar Blue™ cell viability assay. This method can be successfully applied to tumourspheres constructed using low attachment plate and hanging drop plate methods since it is possible to collect cells after the growth / treatment. Bonnier and colleagues reported the way to use the Alamar Blue™ cell viability assay for tumourspheres constructed using hydrogels or scaffold based methods. During this method tumoursphere embedded in gels were incubated with Alamar Blue™ for 24h instead of 3h to get high diffusion of the active dyes through the matrices to cells, similar to our study [24]. On the other hand, the CellTiter-Glo® 3D Cell viability assay is quicker, easier to use and directly applies to the tumourspheres constructed using low attachment plate, hanging drop plate method and scaffold based method.

Ultimately, these basic 3D cell culture models can be further improved to study the role of the blood brain barrier and chemotherapeutic resistance in glioblastoma [31], exploring GBM / normal tissue interactions. The potential impact of the microbiome, TME, vasculature, infiltrating parenchymal and peripheral immune cells on glioblastoma treatment techniques can also be further investigated with more advanced 3D co-culture models [32]. In the future, advances in 3D cell culture will make it possible to generate whole 3D *in vitro* GB organoids, leading to personalized treatments for glioblastoma [20,33,34].

## Supporting information

**S1 File. U-251MG Spheroid generation using low attachment plate method protocol.** Also available on protocols.io. https://www.protocols.io/view/u-251mg-spheroid-generation-using-low-attachment-p-bszmnf46.pdf.
(PDF)

**S2 File. U-251MG spheroid generation using hanging drop method protocol.** Also available on protocols.io. https://www.protocols.io/view/u-251mg-spheroid-generation-using-hanging-drop-met-btstnnen.pdf.
(PDF)

**S3 File. U-251MG Spheroid generation using a scaffold based method protocol.** Also available on protocols.io. https://www.protocols.io/view/u-251mg-spheroid-generation-using-a-scaffold-based-bszqnf5w.pdf.
(PDF)

**S1 Graphical abstract.**
(PNG)

# Acknowledgments

The authors also thank TU Dublin, ESHI, and FOCAS Research Institutes for the use of facilities and support of technical staff.

# Author Contributions

**Conceptualization:** Patrick J. Cullen, Brijesh Tiwari, James F. Curtin.

**Data curation:** James F. Curtin.

**Formal analysis:** Brijesh Tiwari, James F. Curtin.

**Funding acquisition:** Patrick J. Cullen, Brijesh Tiwari, James F. Curtin.

**Investigation:** Janith Wanigasekara, Lara J. Carroll, Patrick J. Cullen, Brijesh Tiwari, James F. Curtin.

**Methodology:** James F. Curtin.

**Project administration:** James F. Curtin.

**Resources:** Brijesh Tiwari.

**Supervision:** Patrick J. Cullen, James F. Curtin.

**Writing – original draft:** Janith Wanigasekara, James F. Curtin.

**Writing – review & editing:** Janith Wanigasekara, Patrick J. Cullen, Brijesh Tiwari, James F. Curtin.

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
