## [Decision Letter · Decision Letter 0]

22 Jun 2022

PONE-D-22-12260Three-Dimensional (3D) in vitro cell culture protocols to enhance glioblastoma researchPLOS ONE

Dear Dr. Curtin,

Thank you for submitting your manuscript to PLOS ONE. After careful consideration, we feel that it has merit but does not fully meet PLOS ONE’s publication criteria as it currently stands. Therefore, we invite you to submit a revised version of the manuscript that addresses the points raised during the review process. To address all points mentioned by the reviewers is mandatory. As you can easily see from the reviews 1 and 3, key critical points were raised that, when dealt with adequately, may add the required relevance and significance. In general, the information provided is not knew and thus, you are very strongly encouraged to improve your work.

We look forward to receiving your revised manuscript.

Kind regards,

Nils Cordes, M.D., Ph.D.

Section Editor

PLOS ONE

Journal Requirements:

2. Thank you for providing the following Protocols.io DOI in the Methods section of your manuscript 

Low attachment plate method - dx.doi.org/10.17504/protocols.io.bszmnf46

Hanging drop plate method - dx.doi.org/10.17504/protocols.io.btstnnen

Scaffold based method - dx.doi.org/10.17504/protocols.io.bszqnf5w

In keeping with our submission requirements, please add the Protocols.io DOI in the “Protocol DOI” field of the submission form (via “Edit Submission”). For more information, please see our submission guidelines: https://journals.plos.org/plosone/s/submission-guidelines#loc-guidelines-for-specific-study-types.

[This study was supported by Science Foundation Ireland (SFI) under Grant Number 17/CDA/4653 and funded through Teagasc Walsh Fellowship. The authors also thank TU Dublin, ESHI, and FOCAS Research Institutes for the use of facilities and support of technical staff.]

 [Science Foundation Ireland (SFI) Grant Number 17/CDA/4653 (BT, PJC, JC)

Teagasc Walsh Fellowship 2017228 (JMW)].

Reviewers' comments:

Reviewer's Responses to Questions

**Comments to the Author**

1. Does the manuscript report a protocol which is of utility to the research community and adds value to the published literature?

Reviewer #1: No

Reviewer #2: Yes

Reviewer #3: Yes

2. Has the protocol been described in sufficient detail?

Descriptions of methods and reagents contained in the step-by-step protocol should be reported in sufficient detail for another researcher to reproduce all experiments and analyses. The protocol should describe the appropriate controls, sample sizes and replication needed to ensure that the data are robust and reproducible.

Reviewer #1: Partly

Reviewer #2: Yes

Reviewer #3: No

3. Does the protocol describe a validated method?

Reviewer #1: Yes

Reviewer #2: Yes

Reviewer #3: No

4. If the manuscript contains new data, have the authors made this data fully available?

Reviewer #1: Yes

Reviewer #2: Yes

Reviewer #3: Yes

**5. Is the article presented in an intelligible fashion and written in standard English?**

Reviewer #1: Yes

Reviewer #2: Yes

Reviewer #3: **No: **There are several grammatical grammatical errors throughout the text, which have to be corrected.

6. Review Comments to the Author

Reviewer #1: Review for PLOS One – PONE-D-22-12260

Wanigasekara et al. “Three-Dimensional (3D) in vitro cell culture protocols to enhance glioblastoma research.”

The manuscript by Wanigasekara et al. with the title “Three-Dimensional (3D) in vitro cell culture protocols to enhance glioblastoma research.” describes the generation of glioblastoma tumorspheroids. Three different approaches are presented, based on the utilization of low attachment plates, hanging drop plates, and cellusponge natural scaffold. Using different cell numbers and incubation times, the authors investigate the growth of these glioblastoma tumorspheroids associated with the three different experimental approaches.

In general, the presented topic touches on an important aspect, as there is a lack of preclinical models recapitulating glioblastoma biology and the associated tumor microenvironment. However, the manuscript is neither comprehensive enough nor sufficiently detailed to support other researchers in the field with the advancement of glioblastoma 3D culture and translational research.

Major concerns:

1. In line with the existing knowledge on molecular heterogeneity of glioblastoma, the investigation of only one cell line is a major drawback of the presented manuscript.

2. The methodological section is partly presented online, however it remains unclear how data were quantified. For example, while at least three experiments were performed, it is not known how many spheroids were measured within each of the three biological repetitions.

3. The manuscript does do not offer new perspectives into how the described 3D models might be of relevance to glioblastoma research. For example: Are these assays adaptable to further functional endpoints? Does the Cellulsponge scaffold mimic physiological architecture of the brain or glioblastoma tumor? etc.

4. The manuscript lacks a critical discussion of the current literature on the topic of glioblastoma 3D culture as well as specific and relevant conclusions on how to move forward in this research field.

Reviewer #2: The manuscript “Three-Dimensional (3D) in vitro cell culture protocols to enhance glioblastoma research” provide a good protocol for different 3D cultures of GBM.

I only miss a discussion, where the author reflects on the data presented from the 3 models. I think a discussion would complement the introduction part and the results.

Reviewer #3: The manuscript by Wanigasekara et al. entitled „Three-Dimensional (3D) in vitro cell culture protocols to enhance glioblastoma research“ describes different ways to generate tumor spheroids from cultured astrocytoma cell lines. This protocol is interesting and helpful per se, however, it is not acceptable in ist current form.

Major points:

1) The protocols were established on one cell line: U-251. This cell line is long-term cultured and used in high passage. As a result, a genetic drift can be anticipated, accompanied by variations in phenotypic marker expression and an increased growth rate in vitro. Particularly, this was previously reported for U-251 cells (Torsvik et al., Cancer Med. 2014 Aug;3(4):812-24). Hence, experiments must be repeated with primary cells freshly taken from glioblastoma patients.

2) The protocol lacks any functional analyses, such as drug response testing (at least towards the most commonly used drug TMZ) or gene expression studies (e.g. changes in expression profiles between individual methods for tumorspheroid formation).

3) The authors describe viability analysis of tumorspheroids using Alamar Blue™ cell viability reagent. The more commonly used method is the CellTiter-Glo® 3D Cell Viability Assay, based on quantitation of the ATP present. I strongly suggest repetition of the analyses using the latter to perform a side-by-side comparison and to draw meaningful conclusions on the method of choice. At least, representative images on viablity, such fluorescence microscopy are warranted to evaluate viability.

4) Experiments on migration/invasion should be incorporated to judge which method is preffered for invasion analyses.

5) This lab protocol lacks information on organoid formation. The authors should at least mention this in vitro model in the introduction part

6) The material & methods section is incomplete, there is no information on the statistics and applied tests. By contrast, in the results section, every third sentence includes ANOVA.

Minor:

7) There are several grammatical errors throughout the text. Please check carefully and correct grammar.

8) Make size information uniform, i.e. 150µm vs. 150 µm.

9) Introduce abbreviation at its first mention in the text, e.g. EMT, ANOVA, etc.

10) Generally, revise the text since there are several typing and spelling errors in the current version.

7. PLOS authors have the option to publish the peer review history of their article (what does this mean?). If published, this will include your full peer review and any attached files.

Reviewer #1: No

Reviewer #2: No

Reviewer #3: No

---

## [Author Response · Author response to Decision Letter 0]

20 Sep 2022

We have included a separate response to reviewers document with this resubmission.

---

## [Decision Letter · Decision Letter 1]

4 Oct 2022

Three-Dimensional (3D) in vitro cell culture protocols to enhance glioblastoma research

PONE-D-22-12260R1

Dear Dr. Curtin,

We’re pleased to inform you that your manuscript has been judged scientifically suitable for publication and will be formally accepted for publication once it meets all outstanding technical requirements.

Kind regards,

Nils Cordes, M.D., Ph.D.

Section Editor

PLOS ONE

Additional Editor Comments (optional):

Reviewers' comments:

Reviewer's Responses to Questions

**Comments to the Author**

1. Does the manuscript report a protocol which is of utility to the research community and adds value to the published literature?

Reviewer #2: Yes

Reviewer #3: Yes

2. Has the protocol been described in sufficient detail?

Descriptions of methods and reagents contained in the step-by-step protocol should be reported in sufficient detail for another researcher to reproduce all experiments and analyses. The protocol should describe the appropriate controls, sample sizes and replication needed to ensure that the data are robust and reproducible.

Reviewer #2: Yes

Reviewer #3: Yes

3. Does the protocol describe a validated method?

Reviewer #2: Yes

Reviewer #3: Yes

4. If the manuscript contains new data, have the authors made this data fully available?

Reviewer #2: N/A

Reviewer #3: Yes

**5. Is the article presented in an intelligible fashion and written in standard English?**

Reviewer #2: Yes

Reviewer #3: Yes

6. Review Comments to the Author

Reviewer #2: I have no further comments and the paper is ready for publication.

I have no further comments and the paper is ready for publication.

Reviewer #3: The authors addressed all my comments and improved the manuscript accordingly. I have no further demands.

7. PLOS authors have the option to publish the peer review history of their article (what does this mean?). If published, this will include your full peer review and any attached files.

Reviewer #2: No

Reviewer #3: **Yes: **Claudia Maletzki

---

## [Editor Report · Acceptance letter]

14 Oct 2022

PONE-D-22-12260R1 

Three-Dimensional (3D) in vitro cell culture protocols to enhance glioblastoma research  

Dear Dr. Curtin:

I'm pleased to inform you that your manuscript has been deemed suitable for publication in PLOS ONE. Congratulations! Your manuscript is now with our production department. 

Kind regards, 

on behalf of

Prof. Dr. Nils Cordes 

Section Editor

PLOS ONE